# Firm-Value Effects of Carbon Emissions and Carbon Disclosures—Evidence from Korea

**DOI:** 10.3390/ijerph182212166

**Published:** 2021-11-19

**Authors:** Jeong-Hwan Lee, Jin-Hyung Cho

**Affiliations:** College of Economics and Finance, Hanyang University, Seoul 04763, Korea; enish27@hanyang.ac.kr

**Keywords:** carbon emission, ESG, CSR, chaebol

## Abstract

We examine the association between carbon emissions, carbon disclosures, and firm value for Korean firms, with a particular interest in *chaebols*, a special type of Korean conglomerate. Using hand-collected carbon emissions and firm-specific data for 841 Korean firms, including 514 *chaebols* and 335 non-*chaebols*, we find a significantly positive relationship between carbon emissions and firm value among *chaebol affiliates.* This result contrasts with previous findings conducted in advanced markets, where investors consider carbon emissions to be destructive. In terms of the voluntary disclosure policy, we find that companies with good environmental performance tend to disclose carbon emissions voluntarily. We further argue that these findings originate from the specific business atmosphere in Korea. Our results support the traditional view of corporations in terms of environmental policy and highlight the importance of firm characteristics and historical developments in the analysis of environmental policy.

## 1. Introduction

Friedman argues that firms should focus on maximizing profits for shareholders, who privately donate their wealth to the causes of their choice [1]. Specifically, he points out that firm managers should refrain from unprofitable behavior regardless of their ownership structure, in accordance with “corporate social responsibility”, which may reduce returns to stockholders. If firms produce unfavorable outcomes to shareholders and the public, investors and the market would attempt to reverse these harmful choices.

With the currently heightened risk from climate change, carbon emission by firms has been brought under scrutiny and public criticism. Consequently, firm managers face greater pressure from several institutions, including the government, media, and NGOs, to *both* reduce and disclose their carbon emissions. For example, environmental organizations, including the Carbon Disclosure Project (CDP) and the Global Reporting Initiative (GRI), have publicly requested firms to increase transparency by disclosing environmental information, such as carbon emissions. Simultaneously, firm managers face growing shareholder pressure to self-evaluate and report the risks and opportunities their firms encounter regarding climate change.

A firm’s reputation for environmental responsibility brings potential benefits to the stakeholder community of the firm. These benefits include revenue growth, positive perception from various stakeholders, including employees, customers, and suppliers, and potential increases in the value of firms [2]. Additionally, pressures from shareholders and various outside organizations serve as a momentum for firms’ internal management control systems to collect information on climate change. Thus, under these pressures, firms would act to disclose their information on carbon emissions while acknowledging that doing so could be disadvantageous to the market.

Several scholars have attempted to address how carbon emissions would drive a redistribution of firm values by pointing out a negative relationship between carbon emissions, their disclosures, and firm value [3,4,5]. However, there have been only a few studies conducted in developing countries where a unique business environment and culture are predominant. In this study, we focus on the relationship between carbon emissions, their disclosures, and firm value for *chaebols*, which are a special type of Korean conglomerate. We hand-collect Korean firms’ carbon emission data from the annual reports of the CDP, a global charity that runs the global carbon emission disclosure system. Our sample period runs from 2013 to 2017 due to the availability of environmental performance data. We use the E(SG) score published by the Korea Corporate Governance Services (KCGS)—an ESG-evaluating institution in Korea—to capture the effect of environmental performance on firms’ tendency to disclose carbon emissions voluntarily. We also employ various firm-specific variables to control for the effects of firm characteristics in the empirical models. 

We chose the Korean financial market to examine the hypotheses for several reasons. The Korean market is ideal for estimating the effect of firm heterogeneity on the determination of ESG policies. Specifically, the Korean financial market has a unique set of family-owned business conglomerations—the *chaebols*. These *chaebols* have played a major role in the nation’s dramatic economic growth with the government’s support. However, over the last decade, critics’ demand for their reformation has grown because of their association with political scandals, which causes owner risk. In this social atmosphere, socially responsible management practices, including environmental protection, have become a major agenda in the demands for both *chaebol* and non-*chaebol* firms [6,7]. While *chaebols* have come under wide criticism and have undergone structural changes, they have faced and continue to meet new “environmental” demands, which we verify and discuss in our analysis.

Unlike other advanced financial markets, it is reasonable to assume that this environmental demand will not yet be reflected in the value of Korean firms, particularly *chaebols*. Rather, the firm value of *chaebols* has been evaluated based on profitability, owing to the long support from the government in the last centuries. Since the 1997 Asian financial crisis, *chaebols* have emerged as extremely profitable firms with less over-investment, despite fewer tax perks [8]. Given the stable profitability of *chaebols* since then, it is reasonable to assume that a substantial amount of carbon emissions could have been a “positive signal” to the market. Larger carbon emissions could imply lower operating costs and a higher rate of return. Therefore, a positive relationship may exist between carbon emissions and firm value, measured as the rate of annual return. As investors pay more attention to the profitability aspect of *chaebol* affiliates, compared to non-*chaebol* affiliates, this positive relationship is probably more significant for the former group of corporations. 

Such value irrelevance or value-enhancing aspects may naturally predict a greater inclination of carbon disclosure among firms with good environmental performance in the Korean financial market. The evaluation of the environmental aspects of a corporation remains at the developmental stage in Korea. Numerous agencies, including the KCGS, have not placed significant weight on the amount of carbon emission in their evaluation [9] but have assigned positive weights to disclosure activity. Consequently, firms that pay considerable attention to environmental performance have strong incentives to disclose carbon emission information voluntarily, to manage their reputations in the Korean market. 

The major findings of our study are two-fold. First, we find a significantly positive relationship between carbon emissions and firm value among *chaebol* affiliates. Non-*chaebol* affiliates do not show a statistically significant relationship between these two variables. This result suggests that the carbon emissions of *chaebol*-affiliated firms are not yet considered value-destructive but favored by the market and investors. This result is in line with the traditional view of corporate value determination [1] but contrasts with most previous research findings [3,5]. Korean investors substantially focus on the cash flow generation aspect of *chaebol*-affiliates, which might reflect their role in the financial market [8]. 

Second, we find a positive relationship between environmental performance and the likelihood of carbon-emissions disclosure. The Korean media and agencies pay limited attention to the amount of carbon disclosure but value the disclosure activities. Consequently, there has been little hesitation to disclose carbon emission information among firms maintaining good reputations in terms of environmental performance. Such a positive association prevails for both *chaebol* and non-*chaebol* affiliates. 

Combining these findings, we find a unique pattern of corporate behavior in terms of carbon emissions for the *chaebol* affiliate group. These firms have positive valuation effects from carbon emissions, potentially due to their influence on cash flow generation. As media and other agencies neglect the amount of emission, these firms do not hesitate to disclose their carbon emissions, regardless of their environmental reputation. 

Our results provide supporting evidence for the traditional theory of corporations [1]. Lowering carbon emissions increases the operating costs of firms and reduces their cash flow generation. Accordingly, a higher level of carbon emissions indicates a greater operating performance, and investors may value such large cash flow generation. This tendency is closely associated with the historical development of the Korean financial market [8]. It adds new insights to the literature, highlighting the difference in ESG policy across advanced and developing financial markets [3,5].

This work is also closely associated with the literature that focuses on the effect of firm heterogeneity on CSR policy determination [10,11,12]. The distinctive environments for *chaebol* affiliates result in significantly different relationships between CSR activities and valuation or disclosure quality across *chaebol* affiliates and non-*chaebol* affiliates [10,11]. Investors in the Korean market value the cash flow generation aspect more highly among *chaebol*-affiliated corporations compared to non-*chaebol* ones; this leads to a positive valuation effect of carbon emissions, only within *chaebol* affiliates. 

The remainder of this paper is organized as follows. Section 2 illustrates previous literature on carbon emissions and firm value, and Section 3 introduces the hypothesis development. Further, Section 4 explains the data and methodology, followed by an analysis of the results in Section 5. Finally, Section 6 concludes the research.

## 2. Literature Review

### 2.1. Climate Change Overview

Greenhouse gas emissions have been identified as the principal cause of climate change over the last century. While there are various greenhouse gases, the share of carbon dioxide (Carbon hereafter), known as CO_2_, is approximately 65% of the total global greenhouse gas emission, surpassing methane (16%) and nitrous oxide (6%). In fact, global carbon emissions from fossil fuels have significantly increased since 1900. Between 1970 to 2011, carbon emissions increased by approximately 90%, with emissions from fossil fuel combustion and industrial processes contributing 78% of the total greenhouse gas emissions [13].

Climate change is due to the exponential increase of this particular pollutant, as it has become a threat to nature and humankind on a global scale. As such, there is a wide consensus that firms should counter the risk of climate change, as it threatens global business [14]. In fact, environmental costs due to climate change can no longer be ignored. For example, Stern argues that if the government does not take preemptive action against climate change, the cost would become 5–20% of the global GDP annually [15].

Among the available carbon emission data, the data from the CDP are regarded as the most reliable information on firm emission. CDP is an international, non-profit organization that aims to help companies and cities disclose their economic impact. It works with over 6000 corporations across over 550 cities and 100 states and regions. The collection of self-reported data from the companies is supported by over 800 institutional investors, with about US$100 trillion in assets.

For firms, participation in the CDP questionnaire was voluntary and on a different set by firms. Firms may respond to or decline participation or may provide partial information, such as links to information generally available on the firm’s website (for example, their CSR/ESG reports), without providing answers to the questionnaires. Finally, firms may respond to the questionnaires and choose strict availability to institutional investors *or* public availability [5].

Although providing carbon emission information to the CDP is not mandatory, the information disclosed through it is regarded as trustworthy [3,5,16]. For firms, the cost of reporting unreliable or false information increases dramatically as they respond to requests from the CDP. Furthermore, the accuracy and reliability of carbon emissions in the market would be enhanced as the interests of stakeholders and the number of reporting firms under the same industry increase, and the validity of carbon emissions through the CDP is confirmed. In fact, it has been reported that firms that previously disclosed carbon emissions through the CDP have a high tendency to disclose consistently through them [17].

### 2.2. Carbon Emission and Firm Value

Although global firms have begun to take initiatives to reduce carbon emissions voluntarily, potential carbon-related costs still follow. For example, firms’ efforts to acquire or develop less carbon-intensive technologies and processes, research and development to produce goods and services associated with low levels of carbon emissions, and other firm initiatives to reduce the carbon footprints of various stakeholders, including employees, would lead to additional cash outflow.

Moreover, a firms’ carbon emissions and disclosures have become an essential element in evaluating potential “un-booked” liabilities and costs along with firms’ financial performances. For example, S&P Global Ratings cut the credit ratings of top U.S. oil producers—Exxon Mobil Corp, Chevron Corp, and ConocoPhillips—by a notch, citing the pressure to tackle climate change as one of the main reasons. As part of heavily carbon-emitting industries, these firms are under pressure from investors and pension funds, which demand that they disclose their carbon footprint and reduce greenhouse gas emissions by investing in renewable energy projects. Although these companies have announced steps to tackle climate change, S&P stated that it does not see them “providing material credit differentiation” [18]. This reveals that non-financial environmental information and performance serve as a framework for assessing firm valuation.

Previous research affirms the negative relationship between carbon emissions and firm value and proves that firms’ voluntary disclosure of their carbon emissions has a positive effect on their firm value. Specifically, observing S&P firm data from 2006 to 2008, Matsumura et al. [3] find that markets penalize all firms for carbon emissions but impose further penalties on firms that do not disclose emission information. Furthermore, they estimate the firm-value effects of voluntary disclosure of carbon emissions, using propensity score matching and doubly regression. Their results reveal that the median firm value is about $2.3 billion higher for firms that disclose carbon emissions compared to firms that choose not to disclose them.

The negative firm-value effect of carbon emissions was further confirmed in studies at international and regional levels. Using carbon emission data from more than 1000 firms in Japan, Saka et al. [19] find that corporate carbon emissions have a negative relationship with the market value of equity, and the disclosure of carbon management has a positive relationship with the market value of equity. They employed a multi-regression model, using the market value and book value of equity, earnings before extraordinary items, and earnings forecast as the main variables. Further, they find that the positive relationship between the voluntary carbon disclosure and the market value of equity is stronger with a larger volume of carbon emissions. After analyzing 5328 observations from 30 emerging countries from 2014 to 2019, Garzón-Jiménez et al. [20] concluded that firms with higher carbon emissions have higher costs of equity, which implies that capital providers penalize polluting firms. Conversely, their analysis indicates that firms with greater environmental disclosures and those who externally ensure their corporate social responsibility reports are met decrease their cost of equity. Hardiyansah et al. [21] used 82 samples of companies listed on the Indonesia Stock Exchange (IDX), which received awards in the Indonesian Sustainability Reporting Award (ISRA) between 2014 and 2018. They noted that carbon disclosure had a positive and significant effect on firm value because it is a form of corporate concern for the environment in response to the market and becomes the basis for investors making their considerations in assessing a company’s sustainability.

Generally, the aforementioned studies point out a negative firm-effect of carbon emissions in particular, regardless of regional location. The main characteristics of these studies on the relationship between carbon emissions, carbon disclosures, and firm value are summarized in Table 1.

### 2.3. Extant Studies in the Korean Market

In 2015, the South Korean government announced its plans to reduce 37% against BAU (business as usual) by 2030. As such, the government has employed various means to reduce carbon emissions: the expansion of nuclear power generation, the commercialization of carbon capture and storage (CCS), the expansion of eco-friendly vehicles, and the introduction of the emissions trading scheme (ETS)—known as the K-ETS. This is the Act on Allocation and Trading of Greenhouse Gas Emissions Allowances and its Enforcement Decree, which passed in 2012, stipulating government actions, institutions, and timelines for the K-ETS.

In alignment with this governmental effort, several Korean firms, including *chaebols*, have actively employed various measures to counter increasing pressure from various stakeholders. In particular, some Korean firms disclose their carbon emissions through the annual report of the CDP. The report discloses the firms’ carbon data and covers various details on a firm’s carbon emissions, such as each firm’s carbon reduction plan, CDP scores, and composition of carbon emissions. In the report, the composition of carbon emissions refers to scope1 (direct emission), scope2 (indirect emission), and scope3 (other indirect emissions not included in scope2).

Previous findings show a negative relationship between carbon emissions and firm value in South Korea. For example, after correcting for selection bias, Choi et al. [5] found a negative relationship between carbon emissions and firm value. Notably, carbon reduction led to an increase in firm value in comparison with the previous year. Using a sample of companies listed in the stock market from 2011 to 2016, Lee et al. [22] analyzed the effect by comparing companies’ respective values (Tobin-Q) and their action viz-a-viz carbon emissions. Their findings reveal that companies that voluntarily reduce their carbon footprints possess higher values than those who had to bow only because of government enforcement. Thus, a CEO’s voluntary environmental awareness has a positive effect on the valuation of a company. However, if firms do not disclose information voluntarily, investors would consider it a reverse signal, regarding it as uncertainty in management.

A branch of literature focuses on the role of *chaebol* affiliates in CSR performance, including environmental activities. Yoon et al. [11] present evidence that they provide well-established financial and non-financial information, which are relatively free from asymmetric information. This is based on the effort and contribution to the overall growth of the Korean economy. Further, Yoon et al. [10] point out that the positive valuation effect of CSR is stronger for *chaebol*-affiliated firms than non-*chaebols*. They find that the valuation effect of corporate governance practice is strongly positive for *chaebols* but negative or insignificant for ordinary Korean firms.

Closely related to our work, there is another branch of literature highlighting the value premium for *chaebol* affiliates. While most studies investigate the evolution of market value in response to a set of corporate policy variations, some studies highlight that the valuation mechanism for *chaebol* affiliates differs from that of non-*chaebol* affiliates [8]. These studies argue that investors value profit-generation ability significantly more for *chaebol* affiliates than for growth options. These works highlight the unique historical background of the Korean market as a key determinant of such different valuations. During the East Asian Financial crisis of 1997, *chaebol* affiliates generated more stable profits compared to non-*chaebol* affiliates due to substantial brand power from economies of scale and a well-functioning internal financing market within a specific *chaebol* group. From the perspective of asset allocation, investors cannot enjoy the benefits of stable and large cash flow generation if they invest in non-*chaebol* affiliates. Thus, they need to focus more intensively on profit generation in the valuation of *chaebol* affiliates.

In terms of environmental policy, the Korean market also has unique features. As the carbon disclosure has not been mandatory, the Korean government and other media have paid limited attention to the potential impact of the absolute carbon emissions of a corporation on the entire environmental system [9]. The hazardous effects of substantial carbon emissions have not been analyzed seriously in government and media reports. However, the evaluation of ESG performance was burgeoning during the 2010s, and the act of voluntary disclosure itself has been considered an environmentally-friendly corporate policy [23]. Unlike other advanced countries, the Korean market is developing, especially regarding environmental policies. We have seen conflicting attitudes across the amount and voluntary carbon disclosure. 

## 3. Hypothesis Development

We now develop an empirical hypothesis to be tested. First, contrary to previous findings, we predict a positive relationship between the value of carbon emissions and firm value in Korean firms, especially for *chaebol*-affiliated firms. The analysis of Lee et al. [8] suggests that the Korean investors apply different criteria for selecting the shares of chaebol and non-chaebol affiliates within their risky asset allocations. Chaebol affiliates are demanded to have stable and large cash flow generations because they need to operate internal financing market properly across affiliates. Such a well-functioning internal financing market is their key competitive advantage over non-chaebol affiliates. However, the Korean investors place greater weight on future growth opportunities rather than currently large profit generation abilities in their valuation of non-chaebol affiliates. 

A higher carbon emission indicates a smaller investment in carbon reduction and lower operating costs, which increases profitability of a corporation. The Korean investors probably favor a large amount of carbon emission within chaebol affiliates because it may imply a higher profitability in the group of firms and their strong internal financing markets. Thus, a positive relationship can be formulated between the degree of carbon emission and a firm’s valuation within chaebol affiliates. 

However, we expect that this positive relationship may not be significant for non-chaebol affiliates. As explained earlier, investors may not significantly value profit generation ability among non-chaebol affiliates. Although a higher level of carbon emission indicates lower operating cost, the investor may not have a high value for profit generation ability compared to chaebol affiliates. Furthermore, contrasting views on the relationship between carbon emissions and firm value [3,5,19] highlight the negative valuation effect of large carbon emissions, especially in terms of future operating costs. Therefore, we expect a less apparent or negative relationship between the amount of carbon emission and firm value within non-chaebol affiliates.

**H-1.** 
**A stronger positive relationship exists between carbon emission and firm value for *chaebol*-affiliated firms.**


The assumption from the first hypothesis naturally leads to the following question: “If markets favor or neutrally value firms for their carbon emissions, would these firms disclose the information voluntarily?” 

We argue that this decision might be more closely related to a firm’s overall management of its environmental policy. If a firm tries to maintain its overall reputation for environmental policy, it has strong incentives to disclose carbon emission information voluntarily in the Korean financial market. While investors may neglect the amount of carbon emission, the media and other rating agencies have paid significant attention to the action of voluntary disclosure [23]. Thus, firms managing their environmental reputation do not hesitate to disclose carbon emissions information. In the face of the positive valuation effect of carbon emissions, disclosing it is optimal to maintain their reputation in these groups of firms.

**H-2.** 
**Firms with high environmental performance would voluntarily disclose their carbon emission information.**


## 4. Data Construction and Empirical Model

In research on carbon emissions, it is essential to use standardized and uniformly comparable information for analysis. As such, we hand-collected data on Korean firms’ carbon emissions from the CDP database.

For firm-specific data, we conducted a subsample analysis based on *chaebol* and non-*chaebol* affiliates. From the FnGuide, we downloaded all data for Korean listed firms between 2013 and 2017 and selected those affiliated with *chaebols*, as defined by the Korean Fair Trade Commission (KFTC), and assigned them to a *chaebol* group. The carbon emission data provided by the KCGS cover a portion of firms listed in the KOSPI market (the largest stock exchange in Korea) due to the availability of carbon emission data. The remaining firms were assigned to the non-*chaebol* group. We estimate the above two empirical models separately for *chaebol* and non-*chaebol* affiliates.

Table 2 provides a summary of Korean firms’ responses to the CDP questionnaires and carbon emissions for the years 2013 to 2017. Of the total sample, 222 firm-years (nearly 27%) disclosed their carbon emissions. However, among those disclosing carbon emissions, *chaebol*-affiliated firms make up 88%, indicating that most *chaebol*-affiliated firms were more willing than non-*chaebols* to disclose their carbon emissions. Although the number of non-chaebols disclosing firms are smaller than that of chaebol-affiliated firms, the proportion of disclosing firms among the non-chaebol affiliates is still more than 8%, which may not cause substantial biases in empirical models with more than 300 samples.

As well, although individual firms in our study lie in different industries, we do not believe this difference would cause significant carbon emission intensities. In fact, Korean industries are largely dependent on manufacturing, which is evident in our study. For example, manufacture-related (Consumer discretionary/industry material/Raw material) industries account for 62% (523/841) of whole industry in our sample.

For H-1, the stock return variable is adjusted for the dividend payments, stock split and M&As. Here, our dependent variable is different from previous research, which used the market value of common equity [3,5]. We believe that our variable better approximates and reflects periodic changes in firm value and profitability during a specific period, at least by reflecting the amount of dividend payments properly. 

We incorporate the model of Matsumura et al. [3] to control for the effect of other firm-specific factors on firm value. In line with their model for carbon emissions and firm value, we employ the following variable as independent variable: CO_2_*_Sales* (Carbon Emission/Sales). We also use the following firm and financial characteristic variables as control variables: *TA_Sales* (Total Assets/Sales), *TL_Sales* (Total Liabilities/Sales), and *NI_Sales* (Net Income/Sales). Each variable’s denominator is the sales of each firm. Additionally, the value of CO_2_*_Sales* was multiplied by 10,000 to match the numerical units of other variables. The definitions of the variables used in the empirical model for H-1 are summarized in Table 3.

As for H-2, we employ binary for the dependent variable, *DISC_*CO_2_, and assign 1 if firms disclose carbon emission information and 0 if not. Next, we introduce independent and control variables to test our hypotheses. We use the environmental score, *ENV*, among the ESG scores from the KCGS as independent variable. Here, *ENV* measures how firms in Korea diligently carries out their environmental responsibility in a particular period. Using the score, we verify whether firms with high environmental scores have a high tendency to disclose their carbon emissions voluntarily. The use of this variable has several implications. Economic theory predicts that firms that are more environmentally proactive through various initiatives have incentives to voluntarily disclose environmental information, including carbon emissions, to reveal their environmental type, which investors and external stakeholders do not directly observe [3]. Therefore, our study attempts to test whether this trend becomes more apparent in the case of *chaebol*-affiliated firms.

Moreover, we use a control variable, *IND_DISC*, defined as the proportion of firms disclosing carbon emissions in affiliated industries to capture both industry pressure and individual optimal disclosure decisions. We also include a variable, *LogTA*, measured using the log of the firm’s total assets. Previous research has reported that firm size is positively correlated with the chance that it will respond [17,24]. We also control for firm growth by including the Return-on-Asset ratio (*ROA*) of the firm to observe profitability, in line with previous findings [3,20].

Consistent with higher-leverage firms providing higher-quality disclosures, we include firm leverage, *Leverage*. As the CDP is a network of large institutional investors, firms with high institutional holdings may tend to disclose their carbon emissions, owing to the investors’ call for more transparent disclosures about socially responsible activities [25]. Finally, we use advertising expenses divided by total assets, *AdvExp*, as a control variable. Regarding the empirical models for H-1 and H-2, the individual data that composed each firm variable, such as *TA_Sales* and *TL_Sales*, were collected from the FnGuide, a data-providing company. We also separated *chaebol* and non-*chaebol* affiliates using the category item published in FnGuide. Finally, carbon emission data for each firm were hand-collected from annual CDP reports, and E(environmental) scores were obtained from the KCGS.

Our empirical model for H-1 can be described as follows. First, in Equation (1), we test the relationship between carbon emissions and firm value, controlling for other firm-specific variables. Here, the firm-specific variables encompass total assets divided by sales *(TA_Sales)*, total liabilities divided by sales *(TL_Sales)*, and net income divided by sales *(NI_Sales)*. We adopt the ordinary least squares (OLS) method to estimate our empirical models in line with Matsumura et al. [3]. The OLS method is widely used in literature because of its robustness. It relies on a limited set of assumptions to obtain consistent estimators. As a firm’s financial variables for a fiscal year are substantially affected by firm-specific events, such as mergers and acquisitions, managerial turnovers, and the outcome of R&D projects, the OLS method is known to be one of the best methods to test empirical hypotheses with financial variables (Equation (1)).
Return = β_0_ + β_1_ CO_2__Sales + β_2_ TA_Sales + β_3_ TL_Sales + β_4_ NI_Sales + ε_it_(1)

The empirical model for H-2 employs a logit model to examine firms’ carbon emission disclosure choices. *DISC_*CO_2_, the dependent variable, is an indicator variable coded as 1 if a firm discloses its carbon emission data to the CDP and allows public disclosure, and 0 otherwise (Equation (2)).
DISC_CO_2_ = β_0_ + β_1_ ENV + β_2_ IND_DISC + β_3_ LogTA + β_4_ ROA + β_5_ leverage + β_6_ AdvExp + ε_it_(2)

Subsequently, we conduct a subsample analysis based on the categories of *chaebol* and non-*chaebol* affiliates to capture the different effects of *chaebol*-affiliated firms.

Table 4 presents the summary statistics of the carbon emission data, along with descriptive statistics for the corporate financial information variables. The table includes the mean, standard deviation, minimum, first quartile, median, third quartile, and minimum and maximum values.

Table 5 presents the correlation coefficients of the variables in the analysis. The table shows that *Return*, the measure of firm value for H-1, is highly correlated with the carbon emissions of firms, as represented by CO_2_*_Sales*. Additionally, the correlation coefficients for *LogTA*, *ENV*, and CO_2_*_Sales* with *DISC_*CO_2_ were 0.36, 0.49, and 0.36, respectively. This indicates that a firm with size and environmental performance has a positive association with a firm’s tendency to disclose carbon emissions. However, the absolute value of the correlation coefficient was still smaller than 0.7; thus, it is not likely to cause multicollinearity problems in the estimations. The absolute values of all the other correlation coefficients were also smaller than 0.7. The correlation is presented graphically in Figure 1.

## 5. Empirical Results

Table 6 reports the estimation results of the firm value model for the entire sample of Korean firms, including both *chaebol* and non-*chaebol* affiliates. To test the effect of carbon emission on firm value, we apply the empirical model in H-1 to the estimation. Specifically, the first column covers all firms, and the second and third columns cover *c**haebols* and non-*chaebols,* respectively. The estimated coefficients and corresponding t-values (in parenthesis) are also reported.

The first column of Table 6 shows a positive relationship between carbon emissions and firm value for *chaebol*-affiliated firms, as specified in the second column. The estimated coefficient was 0.42, which was statistically significant at the 95% level. This significantly positive coefficient indicates the effect of carbon emissions on firm value, which supports H-1 and builds on corporate culture theory. The R^2^ was 0.06, which is low in the absolute term but is considerably consistent with the extant studies. For instance, Matsumura et al. [3] conducted a similar analysis, and their R^2^ values ranged from 0.2 to 0.3.

Our results indicate that markets consider firms’ carbon emissions in *chaebol*-affiliated firms as a positive signal of profitability and value it rather highly. As indicated in the second column, carbon emissions had significantly positive effects on the firm value of *chaebol*-affiliated firms after controlling for other firm-specific variables. This finding is in line with the traditional view of corporate theory [1]. This theory argues for the cost-generating perspective of environmentally-friendly policy and its negative implications on the value of a firm.

Interestingly, this robust result contrasts with the results for non-*chaebol* firms with an estimated effect of −0.04, which is not statistically significant. This result is consistent with the findings of Lee et al. [8]. They argue that the Korean investors mainly consider profitability of chaebol affiliates in their decisions of investments, particularly from the experience of the East Asian Crisis of 1997. Chaebol is a special type of conglomerate operating large internal financing markets across its affiliates. A greater profitability implies the chaebol’s stable and strong internal financing market, which reduces the default probability of all of its affiliates substantially. However, non-chaebol affiliates have limited demand of operating internal financing with currently large cash flow generations. Thus, investors tend value more for the growth options of these firms rather than (current) profitability. In sum, their argument suggests that the valuation mechanism across chaebol and non-chaebol affiliates are different in the Korean financial market

Remarkably, our results contrast with Choi et al. [5], which pointed out a negative relationship between carbon emissions and firm value. This may be due to the fact that we use *Return* as a dependent variable which forecasts the performance of a business, serving as better proxy for profitability. On the other hand, the dependent variable for Choi et al. [5] is market value for all firms, which may exhibit short term instability, plausibly showing negative trend in particular period.

Table 7 reports the estimation results for the carbon disclosure model (H-2). In accordance with H-1, the data encompass the sample of Korean firms, including *chaebol* and non-*chaebol* affiliates. To test the tendency of carbon emission disclosure for firms, we apply a logit model to the estimation. Here, the indicator (dependent) variable is *DISC_*CO_2_, which is coded 1 if the firm voluntarily discloses carbon emission, and 0 otherwise. The estimated coefficients and corresponding p-values (in parenthesis) are also reported.

Interestingly, the relationship between a firm’s environmental performance (*ENV*), as measured in the ESG score, and their tendency to disclose carbon emission was statistically significant at 0.02 across all groups of firms. This finding confirms H-2. As shown in Table 6, excessive carbon emission turns out to be favorable or neutral to the investor. In the face of investors’ perceptions, firms that manage the reputation of environmental policy do not hesitate to disclose information related to carbon emissions, which reinforces their reputation. 

Contrary to all firms and non-*chaebol* firms, *LogTA* and *AdvExp* for the *chaebol*-affiliated firms were statistically significant at 0.55 and 14.10, respectively. This result suggests that firms’ high environmental performance has a positive relationship with their disclosure of carbon emissions, regardless of the form of the firms. The indifference in the coefficient between *chaebol*-affiliated firms and non-*chaebols* might be due to a higher standard with regard to environmental performance. This outweighs the difference in the forms of the firms, which leads to strong transparency in “unfavorable” information on carbon emissions. In particular, this result is in line with the finding of Verrecchia et al. [26] in that firms that are less environmentally damaging are more likely to disclose environmental information voluntarily.

Now, we conduct an additional analysis for H-2 to capture specific effects on carbon emission disclosure. The purpose of this analysis is to reduce or eliminate the endogeneity problem because *ENV* variables in H-2 could be endogenous to firms’ tendency to disclose carbon emissions voluntarily. The issue of endogeneity can be addressed by conducting the following analysis for the subset of the *ENV* variable, which, in turn, proves that the results of H-2 are not due to endogeneity.

For the analysis, we used subsets of the E(SG) score instead of the total environmental score; they are divided into environmental strategy (*Envstrag*), environmental organization (*Envorg*), environmental management (*Envmang*), environmental performance (*Envperf*), and stakeholder action (*stakeholder*). By definition, the disclosure of carbon emissions is most closely associated with the environmental management variable. 

Following the previous analysis, the data encompass a sample of Korean firms, including both *chaebol* and non-*chaebol* affiliates. Owing to the availability of data, we analyzed used data across three years (2013–2015). The new equation for H-2 is as follows Equation (3):DISC_CO_2_ = β_0_ + β_1_ Envstrag + β_2_ Envorg + β_3_ Envmang + β_4_ Envperf + β_5_ Stakeholder + β_6_ IND_DISC + β_7_ LogTA + β_8_ ROA + β_9_ Leverage + β_10_ AdvExp + ε_it_(3)

The detailed regression results are summarized in Table 8. Most importantly, a higher score on the environmental management variable, (*Envmang*), which measures how well a firm organizes decision-making units and plans to invest in relation to eco-friendly management, did not lead to a larger tendency for carbon disclosure. In other words, Table 8 argues that the results in Table 7 are not driven by an endogenous relationship between a higher ESG score and voluntary disclosure. If the endogenous relationship directly matters in the empirical examination, a higher score on environmental management (*Envmang*) should result in a greater tendency to disclose carbon emissions. 

The table also shows the statistical robustness of the *stakeholder* score in the decision of voluntary disclosure for both *chaebol*-affiliated firms and non-*chaebols.* A higher score on the stakeholder indicates that a firm tends to place significant policy weights to maintain reputations from their stakeholders. Their actual policies related to strategy and implementation or their organizational structures are directly captured by other categories of environmental scores. Korean firms striving to earn the trust of stakeholders in terms of their environmental policies tend to disclose information related to carbon emissions. 

This emphasis on the stakeholder score is generally in line with the development of our empirical hypothesis. Reputations from stakeholders are critical to the decision to open carbon emission information to the public. With the expectation of no negative valuation effect from the amount of carbon emission, a firm has considerably strong incentives to disclose information on carbon emissions if the firm tries to maintain its reputation for environmental policies. 

Further, notably, the tendency of *chaebol*-affiliated firms to carbon disclosure was largely dependent on the size of the firms, as evidenced by the robustness in firm size, *LogTA*, which was statistically significant at 0.60. This finding is in line with Table 7 and potentially reflects the increasing level of media attention and evaluation of environmental policy for a larger firm. Under such media pressure, these large firms may “involuntarily” disclose their level of carbon emissions. 

## 6. Discussions

The empirical analysis above for the Korean firms supports the two main empirical hypotheses of this work in general: (1) a significant positive relationship between carbon emissions and firm value for *chaebols* and (2) a significant positive relationship between environmental performance and disclosure of carbon emissions at all firm levels. Notably, the result in H-1 was statistically significant only for *chaebol*-affiliated firms, while the result for H-2 was statistically significantly positive across all groups of firms.

In particular, the two findings suggest that while *chaebol*-affiliated firms are more responsive to the industrial and governmental request of climate risk, they enjoy substantial value premium as their ‘unfavorable’ act of carbon emission is *yet* regarded as strong sign for profitability by market and investors. Simultaneously, *chaebol*-affiliated firms’ act of voluntarily disclosing carbon emission may be due to the significant controlling power of a specific family and its large size, as these firms are under stricter monitoring from the government and receive intensive media attentions for a variety of corporate policies.

In contrast, note that the positive valuation effect of carbon emissions is not widely observed in advanced financial markets [3,19]. The amount of carbon emission generally has a negative effect on the value of firms in these markets, where investors have a deeper understanding of environmental issues. These investors may consider the potential damage to the company brand image from such a large emission or a potentially large operating cost involved in reducing carbon emissions in the near future. However, Korea is a developing market, especially in terms of environmental policy. Thus, such damages to brand image or future costs may not yet be valued as significantly high by investors in reducing carbon emissions under strengthening regulations. This tendency is more apparent for *chaebol* affiliates, where investors have paid extensive attention to accounting profitability.

Unlike the issue of evaluating the effect of carbon emissions, media and other agencies highly value the action of voluntary disclosure itself. As far as the amount of carbon emissions is not value destructive, a firm has strong incentives to open the information of its carbon emissions to the public if it wants to maintain its reputation on environmental policies. As ESG evaluation has burgeoned since the early 2010s in the Korean financial market, environmental performance, especially in terms of reputation among stakeholders, has a positive relationship with the tendency to disclose carbon emissions.

These findings highlight two important aspects of CSR policies. First, our findings emphasize the significance of firm characteristics in shaping the implications of CSR policies. Consistent with the results of Yoon and Lee [10,11,12], our findings show that the unique characteristics of *chaebols* significantly affect corporate policies and even investor perceptions. The *chaebol*-affiliated group has significantly different historical backgrounds and investor relationships throughout the rapid growth of the Korean economy. Our analysis shows that such consideration of heterogeneity is important to the literature on ESG analysis. 

Second, our findings suggest that fast-growing ESG-related policies could be appreciated differently across developing and advanced countries. The Korean market has grown rapidly and extensively and is now sometimes considered an advanced financial market. However, the development of market perception and media understanding is relatively slow in terms of carbon emissions and other environmental issues. Our analysis indicates that Korean investors may underestimate the future costs of large carbon emissions. They may not properly predict the implications of snowballing regulations on carbon emissions, unlike the investors in advanced markets. From now onward, such large carbon emissions have to be reduced significantly by regulations, which incur substantial losses in operating profits. 

Therefore, notably, the aforementioned *two-fold* positive relationship of firm value and carbon disclosure for *chaebol*-affiliated firms could be due to a limited period of observation. Investors in the future may take more environmental concerns into “sustainable” profitability for *chaebols*. In this case, we expect that the positive relationship between carbon emissions and firm value for *chaebol*-affiliated firms could turn around in the near future.

## 7. Conclusions

This study develops two empirical hypotheses using a sample of Korean corporations. We hypothesize (1) a positive relationship between carbon emissions and firm value for *chaebol* affiliates and (2) a positive relationship between environmental performance and carbon disclosure in both firm groups. 

Our empirical analysis confirms these two empirical hypotheses using hand-collected carbon emission data and firm-specific financial statement information. Specifically, for H-1, we found a significantly positive relationship between carbon emissions and firm value exclusively for *chaebol* affiliates. For H-2, the measure of environmental performance is positively related to the likelihood of voluntary carbon emission disclosure for all subgroups in our examination. 

The main contributions of our study are summarized as follows. We find new empirical evidence supporting the traditional view of corporate theory, which highlights the cost-generating aspect of any environmental policy [1]. This finding is closely related to Korea’s unique financial market environment. Korean investors highly value the profit generation aspect of *chaebol* affiliates from historical experience after the East Asian Crisis of 1997 [8]. Furthermore, large carbon emissions could signal a lower cost incurred by environmentally unfriendly policies, which increases the firm’s value. This nature of *chaebols* aligns with a branch of literature that emphasizes the importance of firm heterogeneity in the analysis of the overall ESG policy [10,11].

Furthermore, our analysis suggests distinctive environments related to E(SG) policies across advanced and developing markets [3,5,19,20,21]. Although the Korean market is considered to be one of the rapidly growing and well-established financial markets, almost an advanced one, the environmental policy is still considerably new for investors. Accordingly, the amount of carbon emission has not been strictly regulated, and its economic implications are largely unexamined in the Korean financial market. Our unique finding on the positive relationship between carbon emissions and firm value suggests the importance of the market development stage in the analysis of E(SG) policies. 

Notably, our results could be a temporary one that occurred during the 2010s when the understanding of environmental policy was just initiated. Sustainable development and climate change now lie at the center of the debate in the Korean market as well. As the regulations on carbon emission policy become stricter, investors are expected to penalize the amount of carbon emission substantially in their valuation. The investigation of dynamic changes in the valuation is left for future research.

## Figures and Tables

**Figure 1 ijerph-18-12166-f001:**
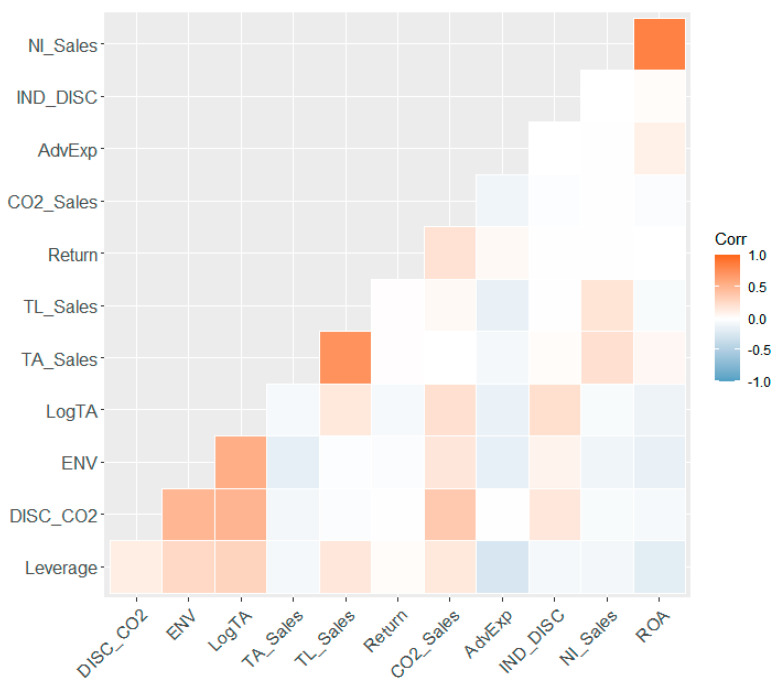
Correlation for variables.

**Table 1 ijerph-18-12166-t001:** Previous research on the relationship between carbon emission, disclosures, and firm value.

Authors (Year)	Firms’ Region	Main Variables	Major Findings/Conclusion
Matsumura et al. (2014) [3]	U.S.	Log of total assets, Ratio of firms disclosing carbon emissions in affiliated industries, Leverage, Foreign sales as a proportion of total sales, EPA’s GHG mandatory reporting rule, Environmentally damaging action, Book to market ratio, Total assets, Total liabilities, Operating profit	Negative firm effects of carbon emission/ further penalty is imposed on firms not disclosing carbon emission
Saka et al. (2014) [19]	Japan	Market value and book value of equity, Residual and net income, Cost of equity capital, Earnings before extraordinary items, Forecast of earnings before extraordinary items	Carbon emissions have a negative relationship with the market value of equity/ Carbon disclosure has a positive relation with the market value of equity
Garzón-Jiménez et al. (2021) [20]	Emerging countries	Cost of equity, Beta, Book to market, ROA, Size, Environmental score, CSR assurance	Higher carbon emissions lead to higher cost of equity
Hardiyansah et al. (2020) [21]	Indonesia	Environmental performance, Profitability, Leverage, Company size, Institutional ownership	Carbon disclosure had a positive and significant effect on firm value

**Table 2 ijerph-18-12166-t002:** Summary of Korean firms’ disclosing CO_2_.

Firm Type	Disclosing CO_2_	Not Disclosing CO_2_
All Firms	222	619
Chaebol	197	317
Non-chaebol	25	310

Note: The number of firms may be partially inconsistent owing to the unavailability of sub-data for some firms.

**Table 3 ijerph-18-12166-t003:** Variable definition.

Variables	Variable Definition	Variable Source
Firm Value Model(H-1)		
Return CO_2__Sales	Adjusted Stock Return Firm’s Carbon Emission/Sales	*FnGuide* *CDP, FnGuide*
TA_Sales	Total Assets/Sales	*FnGuide*
TL_Sales	Total Liabilities/Sales	*FnGuide*
NI_Sales	Net Income/Sales	*FnGuide*
Disclosure-Choice Model (H-2)		
DISC_CO_2_	1 if Carbon Emission Disclosed, 0 Otherwise	*CDP*
ENV	Environment score	*KCGS*
IND_DISC	Proportion of firms disclosing CO_2_ in affiliated industries	*CDP*
LogTA	Natural Logarithm of Total Assets	*FnGuide*
ROA	Firm’s Earnings/Total Assets	*FnGuide*
Leverage	Total Debt/Total Equity	*FnGuide*
AdvExp	Advertising Expenses/Total Assets	*FnGuide*

**Table 4 ijerph-18-12166-t004:** Descriptive statistics.

Variables	Mean	Min	1st Q	Median	3rdQ	Max
Return	0.66	−0.99	−0.13	0.11	1.00	75.4
CO_2__Sales	0.54	0.00	0.00	0.00	0.04	23.65
TA_Sales	1.67	0.32	0.89	1.20	1.71	56.8
TL_Sales	0.73	0.11	0.34	0.51	0.83	11.71
NI_Sales	0.11	−2.06	0.02	0.04	0.08	28.1
DISC_ CO_2_	0.26	0.00	0.00	0.00	1.00	1.00
ENV	144.2	0.00	95.0	156.3	203.0	279.8
IND_DISC	0.26	0.00	0.18	0.23	0.32	1.00
LogTA	21.98	17.81	20.92	21.85	22.97	26.43
ROA	0.05	−0.29	0.01	0.04	0.07	4.08
Leverage	0.19	0.00	0.08	0.19	0.29	0.64
AdvExp	0.01	0.00	0.00	0.00	0.01	0.16

**Table 5 ijerph-18-12166-t005:** Correlation coefficients.

Variables	Return	CO_2__ Sales	TA_ Sales	TL _Sales	NI_ Sales	DISC_ CO_2_	ENV	IND_ DISC	Log TA	ROA	Leverage	AdvExp
Return	1.00											
CO_2__ Sales	0.20	1.00										
TA_ Sales	0.01	0.00	1.00									
TL _Sales	0.01	0.04	0.73	1.00								
NI_ Sales	−0.01	−0.01	0.21	0.18	1.00							
DISC_ CO_2_	−0.01	0.36	−0.08	−0.03	−0.04	1.00						
ENV	−0.03	0.17	−0.17	−0.02	−0.10	0.49	1.00					
IND_ DISC	−0.01	−0.02	0.02	−0.01	0.00	0.16	0.08	1.00				
Log TA	−0.06	0.21	−0.06	0.15	−0.04	0.50	0.55	0.22	1.00			
ROA	0.00	−0.03	0.05	−0.04	0.84	−0.06	−0.14	0.02	−0.12	1.00		
Leverage	0.02	0.15	−0.07	0.16	−0.08	0.12	0.26	−0.07	0.29	−0.19	1.00	
AdvExp	0.04	−0.10	−0.07	−0.14	−0.01	−0.01	−0.15	0.00	−0.13	0.10	−0.26	1.00

**Table 6 ijerph-18-12166-t006:** Results for carbon emission and firm value (H-1).

	Full Sample	Chaebol	Non−Chaebol
Variables	Coeff.	Coeff.	Coeff.
CO_2__Sales	0.31 *** (0.00)	0.42 *** (0.00)	−0.04 (0.51)
TA_Sales	0.01 (0.83)	0.04 (0.88)	−0.01 (0.71)
TL_Sales	−0.02 (0.93)	−0.03 (−0.96)	0.11 (0.43)
NI_Sales	−0.02 (−0.83)	1.07 (0.60)	−0.04 (0.54)
Observations	836	503	327
R^2^	0.04	0.06	0.00
Adjusted R^2^	0.03	0.05	−0.01
F statistics	8.57	8.26	0.307

NOTE: The numbers refer to the estimated coefficients, β. The numbers in parentheses refer to the corresponding *p*-values. *** denotes the 1% significance level.

**Table 7 ijerph-18-12166-t007:** Firm-value effects of decision to disclose carbon emission (H-2).

	All Firms	Chaebol	Non−Chaebol
Variables	Coeff.	Coeff.	Coeff.
ENV	0.02 *** (0.00)	0.02 *** (0.00)	0.02 *** (0.00)
IND_DISC	0.83 (0.18)	1.43 (0.05)	−2.11 (0.26)
LogTA	0.61 *** (0.00)	0.55 *** (0.00)	0.48 * (0.08)
ROA	0.48 (0.66)	0.62 (0.82)	0.89 (0.41)
Leverage	−0.51 (0.54)	−1.69 * (−0.10)	3.57 ** (0.04)
AdvExp	11.99 ** (0.01)	14.10 ** (0.01)	10.17 (0.35)

NOTE: The numbers refer to the estimated coefficients, β. The numbers in parentheses refer to the corresponding *p*-values. ***, **, * denote the 1%, 5%, 10% significance levels respectively.

**Table 8 ijerph-18-12166-t008:** Firm-value effects of decision to disclose carbon emission (H-2) using env. subset.

	Full Sample	Chaebol	Non−Chaebol
Variables	Coeff.	Coeff.	Coeff.
Envstrag	0.02(0.27)	0.01(0.59)	0.06 * (0.10)
Envorg	0.04(0.26)	0.11 ** (0.03)	−0.07(0.37)
Envmang	−0.02(0.14)	−0.03 ** (0.04)	−0.02(0.56)
Envperf	0.03 ** (0.05)	0.03 ** (0.04)	−0.06(0.23)
Stakeholder	0.14 *** (0.00)	0.17 *** (0.00)	0.21 ** (0.01)
IND_DISC	0.35(0.66)	1.00(0.33)	−6.34 ** (0.03)
LogTA	0.55 *** (0.00)	0.60 *** (0.00)	−0.17(0.71)
ROA	0.67(0.73)	−2.12(0.54)	4.38 * (0.06)
Leverage	0.58(0.59)	−0.87(0.51)	5.05 ** (0.03)
AdvExp	9.05(0.12)	16.41 ** (0.02)	−58.35 * (0.08)

NOTE: The numbers refer to the estimated coefficients, β. The numbers in parentheses refer to the corresponding *p*-values. ***, **, * denote the 1%, 5%, 10% significance levels respectively.

## Data Availability

The data that support the findings of this study are available from the corresponding authors upon reasonable request.

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
