# Peer review of "Firm-Value Effects of Carbon Emissions and Carbon Disclosures—Evidence from Korea"

_ijerph, 2021, doi:10.3390/ijerph182212166_

Round 1

Reviewer 1 Report

In this paper, the authors proposed an interesting study concerning the relationship between firm value and carbon emissions as well as carbon disclosures. The research content and the literature review is sufficiently good, so I suggest a minor revision. The main points I am concerned about are as follows for reference in the revision.

  1. How to the environmental performance of firms if they did not disclose CO2
  2. The number of non-chaebol companies disclosing CO2 is much less than other firm types. Please discuss whether this small number of the sample will cause the deviation.
  3. Why did researchers in [5] obtain a different conclusion from yours?
  4. Chaebols and non-chaebols may lie in different industries. Different industries will also have quite different carbon emission intensities, so that affects the relationship between carbon emissions and firm value. Please clarify this.

Author Response

Most of all, we deeply appreciate your precious guidance. We made point by point revisions for each suggestion of yours in the word file. 

Reviewer 2 Report

Dear authors,

I like the paper, in general is well written and interesting.

I just have a minor comment:

I disagree with this sentence: "This result suggests that carbon emissions for non-chaebol firms have limited power to explain the degree of its relationship to firm value". 

However, what are the differences between Chaebol and non-Chaebol firms? Do these differences could influence carbon emissions or on the value of a firm? I think this information is key, so people can understand the study better and it's also important to include as a covariate.
I think the authors should incorporate information about it.

Author Response

Most of all, we deeply appreciate your precious guidance. We made a point by point revision for the suggestion of yours in the word file. 

Reviewer 3 Report

This work conducted a Korean firm-level dataset and examined the firm-value effects of carbon emissions and carbon disclosures. The research was designed properly and well-written, but it was strongly suggested to address the following questions.

  1. this work made a distinction between chaebol-affiliated firms and non-chaebol affiliates. Then, authors should address why the firm-value effects carbon emissions varied across the two types. Meanwhile, I think the H1-hypothesis was not fully demonstrated.
  2. in the firm value model (H1), authors declared the returns was measured by the annual return rate of the adjusted stock price. It was difficult to understand the definition, and the statistic specifications of explaining- and explained- variables was quite different, then it might be the reason for the rough estimations in table 6.
  3. I cannot agree authors’ argument in line 440-441. The R2 of the estimations in table 2 were lower than 10% or 5%, so it might be reasonable to concluded that the estimated model was not properly specified.
  4. authors should give the full definition for the E(SG) indicator.
  5. it was difficult to figure out the association between the two main findings, and authors should give more discussions on the econometric results.
  6. Some key points were indistinct, including environmental management, carbon disclosure, CSR, and so on. It was strongly suggested to integrate the above terms (not to use one word).

Author Response

Most of all, we deeply appreciate your precious guidance. We made a point-by-point response to your valuable comments in the attached word file.

Round 2

Reviewer 3 Report

I agree with your responses and revision. This work was presented well.